# Efficient, cell-type-specific production of flavonols by multiplexed CRISPR activation of a suite of metabolic enzymes

Anaxi Houbaert [1] ✉, Valérie Denervaud Tendon[1], Lukas Hoermayer[1], Nicholas Morffy [2], Lucia C. Strader [2] & Niko Geldner [1] ✉

Synthetic biology in plants promises to transform basic and applied research by rewiring entire developmental modules, signaling cascades or metabolic pathways. Yet, this requires expression of many genes simultaneously, very difficult with classic transgenic approaches, especially for the generation of stable traits. CRISPR activation systems work in plants and could greatly facilitate multiplexed gene activation. Current CRISPR activation systems are efficient for transient or ubiquitous expression. Yet, to fulfill their potential, CRISPR activation needs to perform robustly in specific organs and tissue types. Here, we present a CRISPR activation system that efficiently drives expression in a cell-type-specific manner in stable lines, which requires assessing expression on a cellular basis using fluorescent reporter lines. Our CRISPR systems consistently re-wire gene expression at the cellular level, inducing genes with cell-type specific expression to efficiently express in a new cell layer, such as root endodermis or epidermis. We demonstrate the power of our system to drive functionally relevant, multiplexed gene activation by achieving endodermis-specific production of wild-type levels of flavonoids, detectable by in-situ fluorescence, in a root-flavonoid deficient *myb12* mutant.

A dream of synthetic biologists working on complex multi-cellular organisms is to be able to reach beyond the manipulation of one or a few genes and to re-program entire developmental modules, signaling or metabolic pathway within specific organs or cell types. If this can be achieved in a way that is stably transmitted through the germline, organisms with entirely new, complex traits could be generated. Such manipulations can be achieved by over- or misexpression of transcription factors, which will typically affect expression of hundreds, sometimes thousands of genes, either directly or indirectly, and can cause dramatic alterations in developmental potential[1,2], cellular differentiation[3] or metabolite production[4] in plants. The same evidently applies to animals, where transcription factor over- or misexpression can induce pluripotency[5], homeotic transformations[6] and countless other alterations. Yet, for all their powerful effects, transcription factors are blunt tools that induce cascades of downstream effects and cannot be customized for the activation of precise subset of genes, the way that synthetic biologists envision doing when they conceive complex re-wiring efforts. The advent of Cas9 activation systems carried the promise to allow the concomitant activation of multiple, freely chosen genes at reasonable cost and difficulty. Common to all Cas9 activation systems is the use of a nuclease-dead Cas9 that is either fused to, or interacting with transcriptional activation domains (ADs). This essentially generates an artificial transcription factor whose DNA-binding specificity is determined by the presence of guide RNAs. If guide RNAs are targeted to the promoter regions proximal to the transcriptional start site of a gene, this can cause transcriptional upregulation of a gene with reasonable efficiency. The intriguing feature of such systems is their multiplexing potential, since the simple expression of numerous short guide RNAs, each with different promoter binding specificity, should allow the deliberate

[1]Department of Plant Molecular Biology, Biophore, UNIL-Sorge, University of Lausanne, Lausanne, Switzerland. [2]Biology Department, Duke University, Durham, NC, USA. ✉e-mail: anaxi.houbaert@unil.ch; niko.geldner@unil.ch

activation of any chosen combination of genes. While initial systems, based on direct fusion of Cas9 with activation domains, were shown to work in principle, many improvements were necessary for achieving more reliable and efficient gene activation[7]. Such second/third generation Cas9 activation systems have since been successfully employed for genome wide activation of single genes in cell cultures[8], but also for gene activation in multi-cellular organisms. However, use of Cas9 activation for multiplexed gene activation in this context has remained very limited. In Drosophila, activation of two genes during wing and eye development has been reported, with the enhanced phenotypes compared to single activations suggesting a successful co-expression[9,10], whereas in *C. elegans* modest upregulation of single gene expression of a longevity regulator was reported to lead to significant life span increases[11]. In rats, transient activation of two osteogenic factors in bone-marrow stem cells led to enhanced bone healing when these stem cells were implanted into rats[12]. In mice, stable, germ-line modified lines have been generated that express CRE-lox controlled Cas9 activator system. However, the guide RNAs for multiplexed gene activation were then delivered in a transient fashion into somatic tissues (liver or brain) by viruses[13]. In plants, generation of stable, germ-line transmitted lines is straightforward for a number of model organisms and some important crop species, and numerous CRISPR activation systems have been tested, modified, and compared for their performance[14–17]. In the most recent iterations, such as CRISPR-Combo, two well-performing Cas9 activation systems were combined in order to further enhance activation efficiency, and Cas9 systems allowing simultaneous use for generation of knock-outs, as well as transcriptional activation have been presented[18]. Yet, despite these dynamic developments, little progress has been made in demonstrating the performance of these systems for cell-type specific activation, as well as demonstrating that functionally relevant activation of multiple genes can be achieved in a stable fashion – which is crucial for the use of Cas9 activation for generating plant lines with novel, synthetic traits.

Here we present fluorescent marker-based assays for testing performance of Cas9 activation systems for cell-type specific activation in a developmental context, showing surprising differences in the performance of previously tested systems. We demonstrate performance of our adapted system for cell-type specific activation in several different root cell types. Importantly, we demonstrate that strict testing protocols are required for achieving the synthetic activation of a complex trait, such as a metabolic pathway, in which each gene must be activated to functionally relevant levels to achieve synthesis of an end-product. Using these testing strategies, we achieve the stable activation of a full metabolic pathway in a specific cell-type fashion, involving the activation of up to 6 genes concomitantly, demonstrating that CRISPR activation technology can be used for the synthetic generation of complex, stable traits in multi-cellular organisms.

## Results

In order to test and benchmark different CRISPR activation systems for their ability to efficiently drive cell type specific expression, we needed to establish fluorescent transcriptional readouts, since standard qPCR methods do not have the required spatial resolution and thus do not allow to assess whether an efficient gene activation is due to low expression in many cells or high expression in a restricted subset of cells, for example. We decided to use endodermis-specific activation as our test system, since this primary root cell layer is of central importance for root function, and fluorescent read-outs can be easily observed. As endodermis-specific promoter driving the Cas9 activation systems, we chose the promoters of LTPG20 as well as PER03, both driving strong and specific expression, with PER03 displaying earlier, meristematic expression, compared to LTPG20 (Supplementary Fig. 1). We chose three popular Cas9 activation systems, previously shown to work in plants[14–16], dCas9-Suntag, dCas9-TV and dCas9-

Act2.0. The dCas9-Suntag system is a two-component system in which the nuclease-deficient ("d") Cas9 is fused to a "tail" of 10 concatemerized sequences of the short GCN4 antibody epitope. These short peptide sequences are bound by single chain fragment variable (scFv) recombinant antibodies, fused to a superfolder GFP and a VP64 (4xVP16), theoretically allowing for up-to ten activation domains being recruited to one dCas9 unit. Because of the need for fluorescent transcriptional read-outs, we eliminated the bulky superfolder GFP portion from the scFV component, thus generating a smaller scFv-VP64 unit, potentially advantageous for binding of multiple units to the GCN4 tail (Fig. 1a). dCas9-TV systems and dCas9-Act2.0 systems were left unmodified, except for their placement in a new vector backbone, under the control of the LTPG20 promoter. Additionally, the modified gRNA scaffold containing the MS2 aptamer used for dCas9-Act2.0 was introduced into our gRNA expressing vectors containing the promoters AtU6-26 and AtU3, instead of AtU6-1. Three different transcriptional fluorescent reporters were chosen for assessing activation efficiency in the endodermis−pLOVE1, expressing exclusively in root cap cells, as well as pGPAT2 and pGPAT3, expressing exclusively in the epidermis and root cap. Their restricted, cell-type specific endogenous expression patterns allowed for easy observation of an ectopic endodermal expression, as well as a scoring of any possible activation outside of the endodermis. In each case, nuclear-localised fluorescent reporter proteins allowed for unambiguous assignment of signals to a specific cell type. Three guide RNAs were used for activation of each reporter gene (Fig. 1b). For an unbiased comparison of the performance of the three dCas9 systems for endodermis-specific activation, we performed a T1 analysis, scoring a minimum of 77 independent lines and categorized their activation strength from whole root fluorescent images (Supplementary Fig. 2). The large number of T1s used allowed to eliminate bias due to the random insertion position of the transgene associated with Agrobacterium-mediated transformation. Surprisingly, the modified Suntag system clearly outperformed the other two systems in activation potential, both for activation of pLOVE1 (Supplementary Fig. 2a–h) and pGPAT2 (Supplementary Fig. 2i–p) promoters. Activation was highly specific to the endodermis in most independent T1 (Fig. 1c–e) with a "per cell" expression strength equal to or stronger than the expression in the endogenous cell types.

With the aim of demonstrating the broader utility of the Suntag system for activation in other cell-types, we placed the two components of the system under the control of the epidermis-specific GPAT3 promoter and the cortex-specific PEP promoter. Additionally, we checked its performance under the ubiquitous UBQ10 promoter, as well as the endodermis-specific promoter PER03, driving earlier endodermal expression in undifferentiated cells (Fig. 2). Again, we used pLOVE1 promoter activation as a read-out since its restricted expression in root cap cells allowed for easy assessment of activation efficiency for all of the four additional promoters. In all cases, lines with strong and consistent activation in the cell types of interest could be achieved, demonstrating the robustness and portability of this activation system (Fig. 2). It is important to note, however, that significant line-to-line variability exists (Supplementary Fig. 3a, b) and that for the ubiquitin and GPAT3 promoter system, surprising root-or-shoot specific silencing could be observed (Supplementary Fig. 3c). Again, such complex differences between lines can only be detected by using fluorescent markers as activation read-outs, as we are doing here. With standard qPCR methods, such lines would have been simply categorized as being of intermediate strength. Therefore, pre-testing and selection of specific, well-performing CRISPR activation lines is crucial for a reliable gene activation performance.

In an attempt to further optimize the Suntag system, we replaced the multimerized animal virus activation domain VP64 with a 2xTAL-ACTIVATION DOMAIN (2xTAD), the activation domain of an effector of a plant pathogenic bacterium[15]. Both activation domains worked

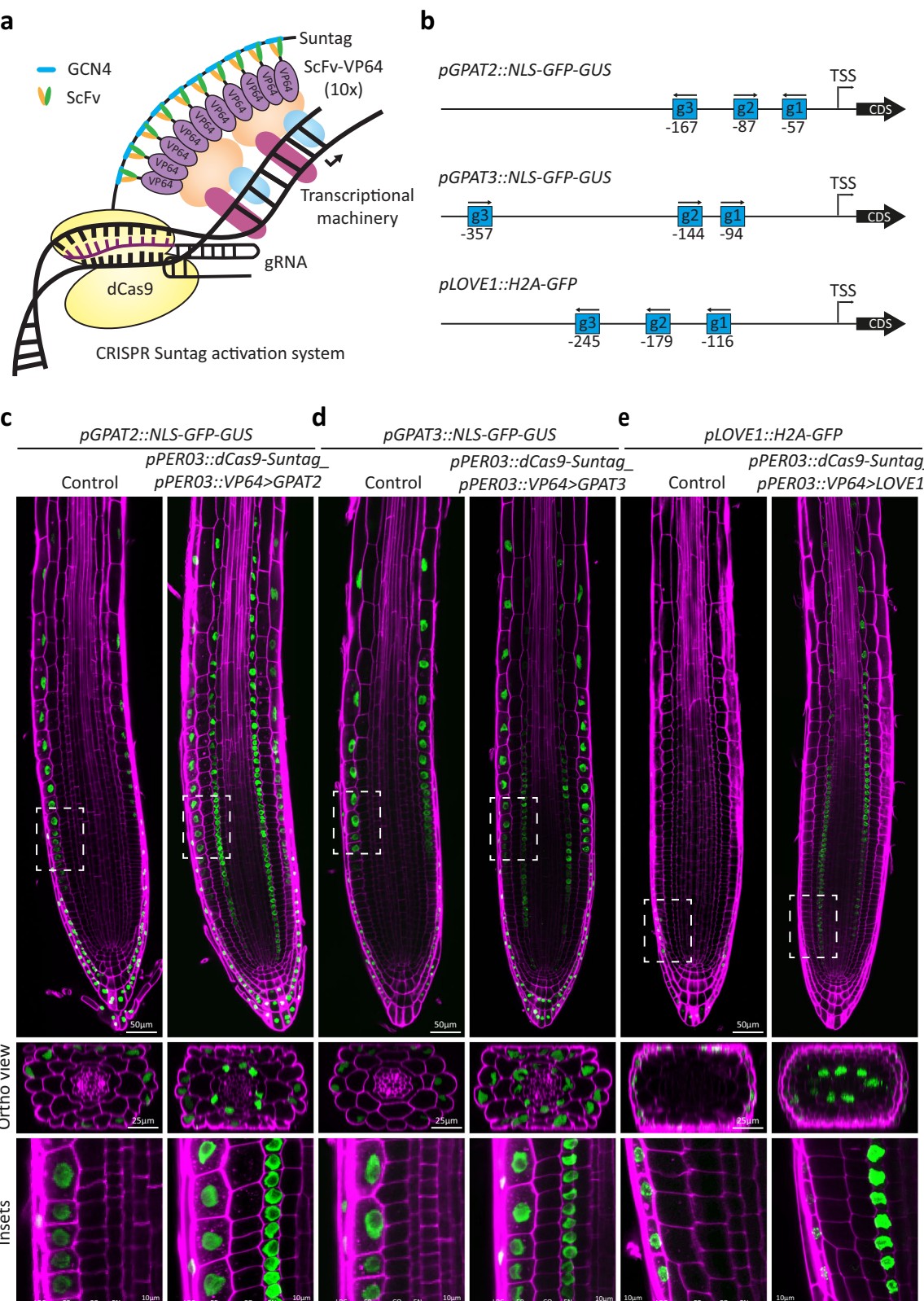

reliably (Supplementary Fig. 4a), although VP64 clearly outperformed 2xTAD with respect to highest expression potential (Supplementary Fig. 4b, c). However, 2xTAD, while showing less maximum activation potential, showed more lines displaying an at least "fair activation" (Supplementary Fig. 4c). Thus, 2xTAD can be seen as more reliable when at least some degree of activation is desired in a high percentage of non-selected lines. qPCR analysis of reporter gene activation in

pooled T1 CRISPR activation lines under different promoters supported this conclusion, with a consistently higher average activation observed for VP64 containing lines (Supplementary Fig. 4d). This correlated rather well with the higher average expression of both components of the activation system in the VP64 variants of the system (Supplementary Fig. 4e). This better transcript accumulation is surprising, considering that both systems are expressed from the same

**Fig. 1 | Endodermis specific transcriptional activation using CRISPRa Suntag.**
**a** Schematic representation of the dCas9-Suntag activation system. The nuclease dead Cas9 (dCas9) is fused to the Suntag epitope tail containing 10 repeats of the GCN4 epitope. The single chain antibody fragment (ScFv) recognizing the GCN4 epitope is fused to the VP64 activation domain. The superfolder GFP was removed from the original vector (Papikian et al., 2019) to allow transcriptional reporters to be used as visual readouts. **b** Schematic representation showing the gRNAs designed to target the promoter regions of GPAT3, GPAT2, and LOVE1. The distance from the transcriptional start site (TSS) is presented under the gRNA. **c** Endodermis

specific transcriptional gene activation of the reporter for GPAT2 (**c**), GPAT3 (**d**), and LOVE1 (**e**) in 5-day-old seedlings of a stable lines expressing *pGPAT2::NLS-GFP-GUS*, *pGPAT3::NLS-GFP-GUS*, or *pLOVE1::H2A-GFP*, respectively. The dCas9-Suntag activation system was specifically expressed in the endodermis using the *PERO3* promoter (*pPERO3::dCas9-Suntag-pPERO3:: ScFv-VP64*). Seedlings were fixed with paraformaldehyde (PFA) and stained with calcofluor white (CW) to visualize the cell walls. Insets are displayed by dashed rectangles. These experiments were repeated twice with similar results. Scale bars 50 μm (root tip), 25 μm (orthogonal view), or 10 μm (insets). LRC lateral root cap, EP epidermis, CO cortex, EN endodermis.

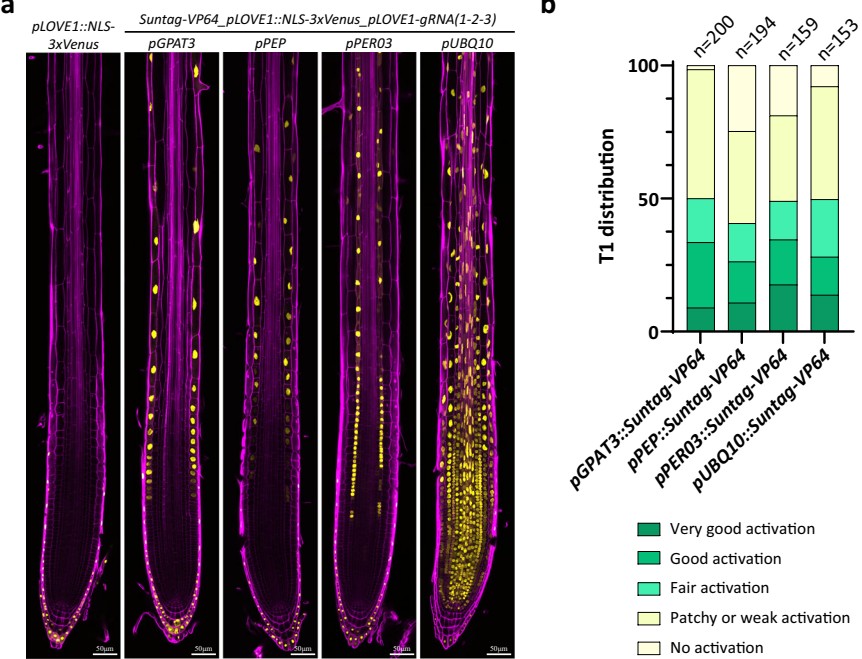

**Fig. 2 | Tissue specific transcriptional activation in the root of *Arabidopsis* seedlings. a** Tissue specific transcriptional activation of the LOVE1 reporter *pLOVE1::NLS-3xmVenus* in roots of 5-day-old seedlings. LOVE1 is normally expressed specifically in the root cap cells. The dCas9-Suntag activation system was expressed under the GPAT3 (*pGPAT3::dCas9-Suntag pGPAT3::ScFv-VP64*), PEP (*pPEP::dCas9-Suntag pPEP::ScFv-VP64*), PERO3 (*pPERO3::dCas9-Suntag pPERO3::ScFv-VP64*), and UBQ10 (*pUBQ10::dCas9-Suntag pZmUbi::ScFv-VP64*) promoters to activate the transcription of the reporter of LOVE1 in the epidermis, cortex, endodermis, or

whole roots, respectively. **b** The distribution of independent T1 lines. Individual seedlings were scored based on the activation strength and stability of the *pLOVE1::NLS-3xmVenus* reporter in different tissues. The seedlings were arbitrarily attributed to one of the five groups ("Very good activation", "Good activation", "Fair activation", "Patchy or weak activation", and "No activation"). Seedlings were fixed with PFA and stained with CW to visualize the cell walls. Scale bars 50 μm. *n* = number of seedlings analyzed.

promoter and in the same vector context. We next tested 15 additional ADs (Supplementary Fig. 4f-k, Supplementary Data 1), recently identified in an activation screen in yeast aiming at mapping Arabidopsis transcriptional activation domains[19]. The VP64 AD was replaced by a tandem copy of each of the 15 selected ADs, mimicking the design for 2xTAD. Out of the 15 new activation systems, AD1, AD2, and AD9 could reliably activate gene transcription to levels comparable to that of VP64, making these plant activation domains potentially interesting alternatives to the widely used viral VP64 domain in future iterations of Cas9 activation systems.

The promise of CRISPR activation systems resides in their ability for multiplexed gene activation using small guide RNA arrays, and thus allow for a complexity of gene activation that goes beyond what can be achieved with transgenic technology at reasonable effort and cost. We therefore set out to test whether we could achieve functionally relevant activation of a larger number of genes using our system. Biosynthetic pathways of secondary metabolites are ideal for such a purpose since metabolite production depends on the concomitant activity of every enzyme in a pathway. The pathway for flavonol production is very well understood in plants[20], with loss-of-function

mutants of enzymatic genes leading to identical phenotypes[21], demonstrating the requirement of each single enzyme for the functioning of the pathway as a whole. The pathway consists of 5-6 enzymes, together driving production of the flavonols kaempferol and quercetin (Fig. 3a). The pathway branches off from p-coumarate, produced from the widely present pathway for phenylpropanoids, required for production of lignin monomers and many other products. The first enzymes of the phenylpropanoid pathway, phenylammonium lyase (PAL) and cinnamate-4-hydroxylase (C4H), are known to be expressed in the endodermis (Fig. 3c), making this cell type a good candidate for the generation of flavonol production by specific transcriptional induction of the downstream enzymes. Wild-type Arabidopsis plants produce significant amounts of flavonols in their roots[22-25], which would make it difficult to reliably assess CRISPR activation-induced flavonol production in the endodermis. Fortunately, flavonol production in roots largely depends on a single MYB transcription factor, MYB12, leading to strong reduction or absence of expression of most of the flavonol biosynthetic enzymes in root of the *myb12* single mutant (Fig. 3b, Supplementary Fig. 5a) and, consequently, barely detectable levels of kaempferol and quercetin in most

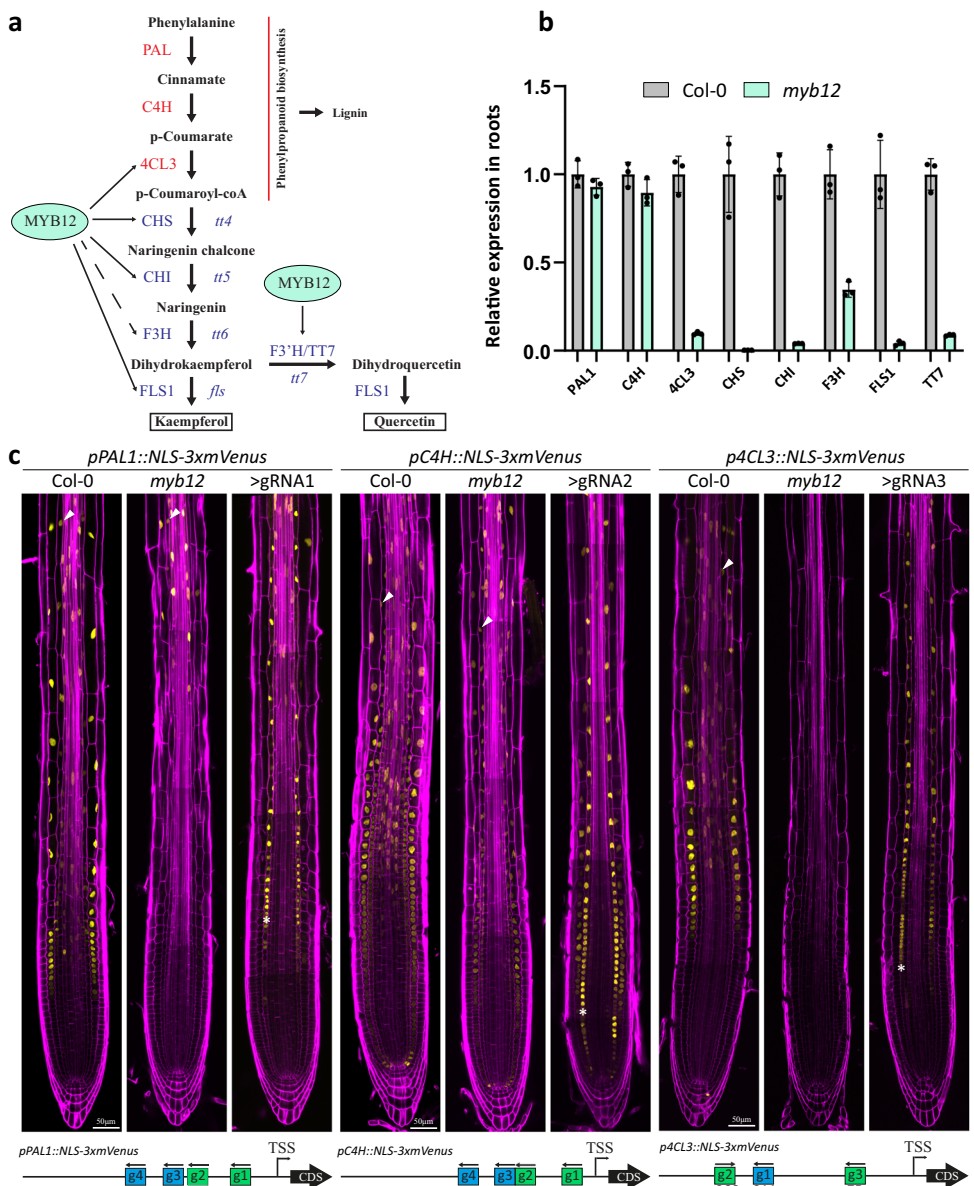

**Fig. 3 | Transcriptional activation of the flavonol pathway in *Arabidopsis* roots.**
**a** The flavonol biosynthetic pathway. Arrows emanating from MYB12 display transcriptional regulation. The dashed arrow emanating from MYB12 displays partial transcriptional regulation. Enzymes colored in red are involved in the phenylpropanoid pathway. **b** Relative expression of the biosynthetic enzymes PAL1, C4H, 4CL3, CHS, CHI, F3H, FLS1, and TT7 in roots of pooled 5-days-old seedlings of Col-0 and *myb12* measured by qPCR. MYB12 regulates the expression of 4CL3, CHS, CHI, FLS1, TT7, and partially of F3H. The graph displays a representative experiment from 3 independent biological repeats. Data are presented as mean values, and the error bars indicate the standard deviations (SD). **c** Transcriptional reporters of *pPAL1::NLS-3xmVenus*, *pC4H::NLS-3xmVenus*, and of *p4CL3::NLS-3xmVenus* in Col-0

and *myb12*. PAL1 and C4H are still expressed in the root endodermis of *myb12* as indicated by white arrowheads. 4CL3 is expressed in the endodermis in Col-0 but is no longer expressed there in *myb12*. gRNA activation potential in the endodermis of *myb12* was scored for PAL1, C4H, and 4CL3 by expressing the dCas9-Suntag-2xTAD activation system in the endodermis using the PERO3 promoter. The gRNA positions are represented on the schematics under the picture. gRNAs in green represent gRNAs with the best activation potential. The best working gRNA was selected for imaging. Transcriptional activation in the endodermis is indicated with a white asterisk. Seedlings were fixed with PFA and stained with CW to visualize the cell walls. Scale bars 50 μm.

parts of the root[23]. We therefore used *myb12* as the background for our attempt at driving endodermis-specific flavonol production by CRISPR activation-mediated pathway activation. To do so, we used our *pPERO3::dCas9-Suntag pPERO3::ScFv-VP64* version of the activation system and assembled it with two arrays of 6 guide RNAs into a single vector (Supplementary Fig. 5b). Each of the six flavonol biosynthetic enzyme genes was targeted by two guide RNAs, selected according to previously established prediction criteria[26,27]. However, among the 17 T2 lines generated and characterized, none showed any flavonol accumulation in roots. Characterization of these lines revealed the

reasons for this failure. Firstly, an important line-to-line variability in the expression of both components of the CRISPR activation system was observed (Supplementary Fig. 5c). The observed differences in expression strength predicted quite well the performance of the system in activating the flavonol biosynthetic genes, assessed by qPCR (Supplementary Fig. 5d). More critically however, we observed that, even in lines in which the Cas9 system is well expressed, some genes showed poor or no activation, most probably due to poor performance of the predicted guide RNAs. Based on these observations, we decided to optimize both expression of the Cas9 activation system and

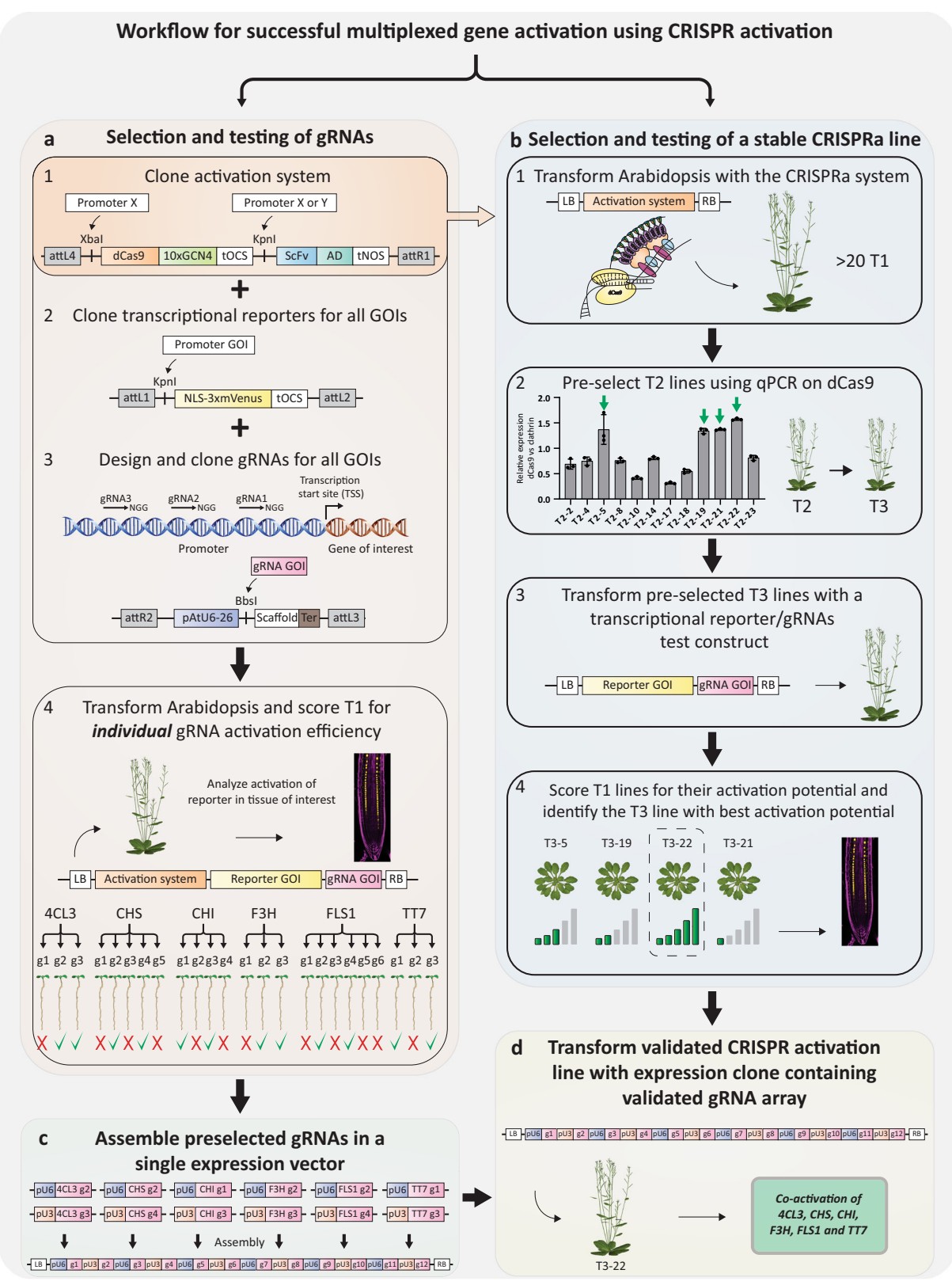

**Workflow for successful multiplexed gene activation using CRISPR activation**

guide RNA performance. To do so, we pursued a two-pronged selection system. For guide RNA selection, we assembled a vector with the Cas9 activation system in the first position, the transcriptional reporter of the gene-of-interest (one of the flavonol biosynthetic gene promoters) in second position and an expression cassette for a single guide RNA in third position (Supplementary Fig. 6a). At least three individual guide RNAs were used per promoter, leading to 24 vectors

that were transformed and tested for their activation potential in analysis of T1 lines (Fig. 4a). With this method, we identified two guides for each gene that performed well individually on the promoter::reporter construct (Fig. 3c, Supplementary Fig. 7a, b). These were then used to be assembled into a 12 guide RNA array of validated guides (Fig. 4c, Supplementary Fig. 6b). We compared this relatively labor-intensive strategy of testing guides in stably transformed seedlings

**Fig. 4 | Workflow for successful multiplexed gene activation using CRISPR activation.** Schematic representation of the workflow for optimal gene activation in Arabidopsis. A and B are executed in parallel. **a** Cloning strategy for the selection and testing of gRNAs. 1) Tissue specific promoters are cloned in front of the dCas9-10xGCN4 Suntag component using XbaI and in front of the ScFv-AD (activation domain) using KpnI to generate a cell type specific activation system (Gateway entry clone L4R1). 2) The promoter of the genes of interest (GOIs) is cloned in front of the NLS-3xmVenus cassette using KpnI to generate transcriptional reporters (Gateway entry clones L1L2). 3) Multiple gRNAs are designed and tested individually. A single gRNA (sgRNA) targeting the promoter region of the gene of interest is cloned into an entry clone using BbsI and oligo annealing (Gateway entry clones R2L3). The entry clones are recombined using an LR reaction and transformed into the genetic background of interest. Following transformation, 20 individual T1 seedlings are selected to assess their gRNA activation potential. **b** Workflow describing the steps suggested for establishing a stable activation line. 1)

Transformation of the activation system under control of tissue specific promoters (obtained in A1) in the genetic background of interest. 2) Characterization of >20 T2 lines by qPCR to assess the expression levels of the dCas9. It is important to note that a high expression of the activation system does not always correlate with a high transcriptional activation potential. It is thus essential to select multiple lines to bring to homozygous T3 generation. The graph displays a representative experiment from 3 independent biological repeats. Data are presented as mean values, and the error bars indicate the standard deviations (SD). 3) Transformation of the preselected homozygous lines with a vector expressing a transcriptional reporter and the gRNAs targeting that promoter (In this example *pLOVE::NLS-3xmVenus_pU3/U6::pLOVE1gRNAs 1,2,3*) 4) Score the lines by assessing the activation strength in 20 T1 seedlings. Continue with the best stable activation line. **c** Assemble the preselected gRNAs targeting the GOIs using golden gate and gateway cloning into a single expression vector. **d** Transform the stable activation line with the stacked gRNAs expression vector.

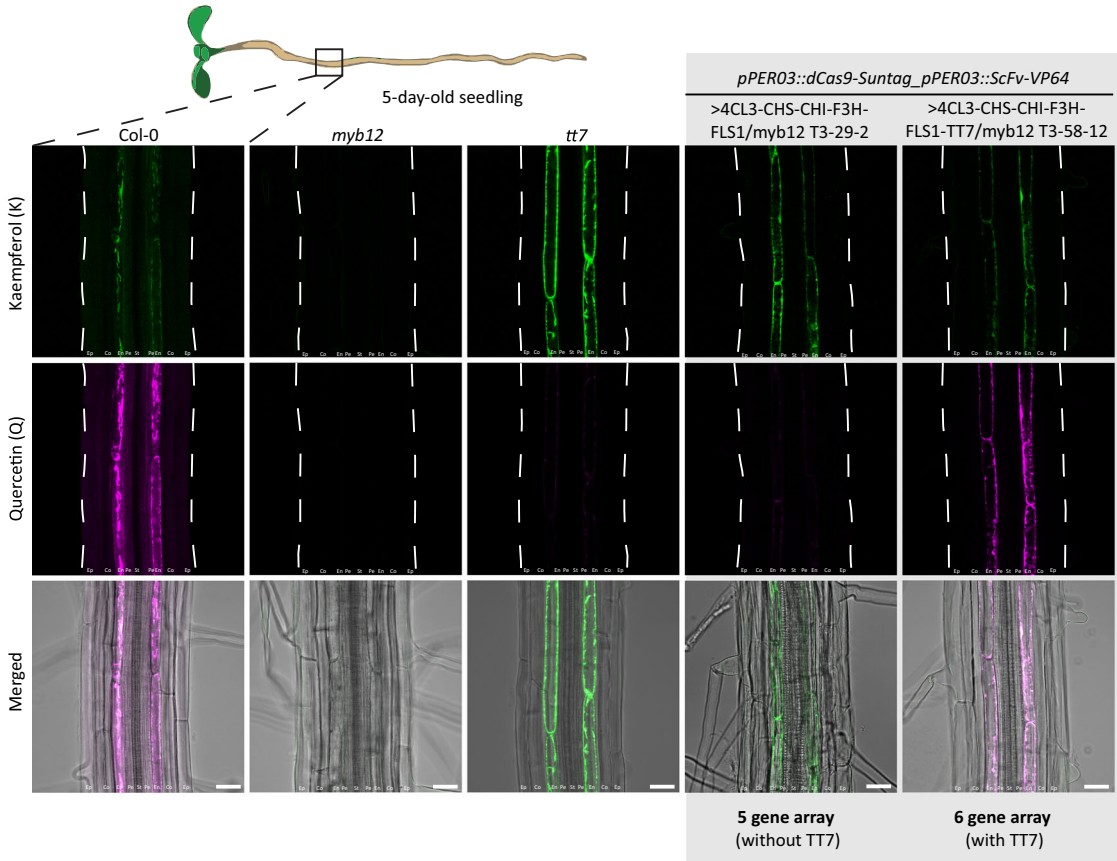

**Fig. 5 | Reconstitution of the flavonol biosynthetic pathway in the endodermis of *myb12*.** Production of kaempferol and quercetin was reconstituted specifically in the *myb12* endodermis in roots of 5-day-old seedlings by co-activation of 4CL3, CHS, CHI, F3H, FLS1 and TT7 (+TT7). Co-activation of 4CL3, CHS, CHI, F3H, FLS1 without TT7 (-TT7) leads to production of just kaempferol. The *tt7* mutant devoid of

quercetin and over-accumulating kaempferol was used as a control. The schematic representation of the seedling shows the area that was imaged. These experiments were repeated twice with similar results. Ep Epidermis, Co Cortex, En Endodermis, Pe Pericycle, St Stele. Scale bars 25 μm. The contours of the roots are shown with a dashed white line.

with results we obtained with assays using transient protoplast expression, commonly used for guide RNA testing[15,27]. While roughly comparable, we observed some clear differences, with guide RNAs performing well in protoplast assays, failing, or performing more badly in a stable context (Supplementary Fig. 7b, c).

In parallel, we used vectors containing the Cas9 activation system only and generated a large number of T2s, which were pre-selected for good Cas9 expression and subsequent establishment of homozygous T3 lines (Fig. 4b, Supplementary Fig. 8a). These lines were then additionally tested for their actual activation potential by super-

transforming a pLOVE1 reporter guide RNA vector into them and assessing their activation potential in T1 analysis. Untransformed plants of the best performing T3 line were then used to introduce the validated guide RNA array (Fig. 4c, d).

By using this two-pronged validation strategy, we were indeed able to obtain lines that reliably activated all six enzyme genes (Supplementary Fig. 8b) and induced wild-type levels of flavonol production in the endodermis (Fig. 5, Supplementary Fig. 9a–d). We thus achieved a functional, concomitant activation of six genes by Cas9 activation. Detecting flavonol production

with cell-type resolution is possible by using an established in-situ DPBA staining method, which provides a spectrally distinct fluorescence to both kaempferol and quercetin[24,28]. Interestingly, beyond the previously described accumulation of flavonols in the root transition/elongation zone, the vasculature/pericycle and young lateral roots of wild-type plants, we also observed a very strong signal in mature endodermal cells with this method. The presence of flavonols in this tissue was corroborated by the strong abundance of the biosynthetic enzyme FLS1 in mature endodermis (Supplementary Fig. 9e). Expectedly, flavonols are mostly absent in the roots of 5 days old *myb12* mutant, except for a very weak endodermal signal around emerging lateral roots (Fig. 5, Supplementary Fig. 9a, c). Further validating the specificity of the staining method for kaempferol and quercetin, the red quercetin signal is absent in a *tt7* mutant, with an increase in the green signal, consistent with an increase in kaempferol, the TT7 enzyme substrate and precursor of quercetin (Fig. 5, Supplementary Figs. 8c, 9a–d). In perfect accordance with this, CRISPR activation lines transformed with a 10 guide RNA/5 gene array that does not include TT7 displayed exclusively green fluorescence in the endodermis, indicating the exclusive accumulation of kaempferol, while a line transformed with a 12 guide/6 gene array, including TT7, displayed only weak kaempferol and strong quercetin accumulation (Fig. 5, Supplementary Fig 9a–d). Despite the successful endodermis-specific production and accumulation of flavonols in our activation lines, we observed a significant delay of flavonol accumulation in the endodermis, compared to the onset of PER03 promoter activity. Moreover, we did observe occasional, weak, but broad accumulation of flavonols in emerging lateral root meristems. In order to understand whether this is due to delayed or ectopic activity of the dCas9 system, we generated a line directly expressing MYB12 under the control of the same endodermal promoter in the *myb12* mutant background. PER03 overexpresses MYB12 and leads to strong accumulation of flavonols in the endodermis, whilst rescuing flavonol accumulation in the pericycle and stele of Arabidopsis roots (Supplementary Fig. 9f, g). Interestingly, even in this line, a significant delay in flavonol accumulation in the endodermis is observed (Supplementary Fig. 9f). Moreover, pPER03::MYB12 also shows a pronounced, broad accumulation of flavonols in the primary and lateral root meristems (Supplementary Fig. 9f). We interpret this as an indication of flavonol transport, allowing accumulation of this metabolite outside of its site of production. Indeed, when comparing the wild-type expression pattern of flavonol biosynthetic enzyme genes (Fig. 3c, Supplementary Fig. 7a) and the flavonol accumulation pattern (Fig. 5, Supplementary Fig. 9a–d), a similar disconnect between enzyme expression and metabolite accumulation is observed, especially in cortical cells.

We were also able to obtain flavonol accumulation when we expressed the same guide RNA array in combination with the CRISPR activation system driven under the lateral root cap/epidermis-specific GPAT3 promoter. Here, weak flavonol accumulation in primary and stronger accumulation in lateral root meristems was observed, with weak and sporadic accumulation in endodermis or stele (Supplementary Fig. 9h). Again, the accumulation was extending beyond the cell types in which GPAT3 is active, suggesting active metabolite transport. Corroborating this, a line overexpressing MYB12 directly by the GPAT3 promoter in the *myb12* mutant background (Supplementary Fig. 9g) displayed a stronger but similarly extended accumulation of flavonols in primary and lateral root meristems (Supplementary Fig. 9i). Intercellular transport processes thus appear to play an important role in determining flavonol accumulation patterns. Indeed, numerous transporters have been identified that drive both intracellular and intercellular flavonol transport[29–31].

## Discussion

### dCas9 systems for stable gene activation in a cell-type specific, developmental context

Our comprehensive comparison of the performances of different, recent Cas9 activation systems for cell-type specific activation reveals a much more reliable performance of our simplified Suntag system compared to Act2.0 for cell-type specific activation in roots. Our work indicates that performance of Cas9 activation systems in one specific context cannot predict their performance when different promoters, vector backbones, and read-outs are used. We suggest that researchers interested in using Cas9 activation systems should test a number of different systems when transferring them into different plant species and aim for activation in different organs or cell types. Although we did not compare the performance of our simplified Suntag activation system devoid of the superfolder GFP in the ScFv module with the original Suntag[14], other groups have shown that removing the GFP vastly improved the activation potential of the system, probably by reducing sterical hindrance[32]. In an attempt to further improve the Suntag activation system, we compared the activation domains VP64, 2xTAD and 15 others identified in Arabidopsis within the context of cell-type specific transcriptional activation. While others reported that 2xTAD outperformed VP64 when activating OsER1 in rice protoplasts[27], we observed a lower activation potential of 2xTAD on average in stable *Arabidopsis* seedlings when activating the pLOVE1 reporter. However, 2xTAD lines displayed a more robust activation, with less cell-to-cell variability. It is worth noting that the CRISPRa-Act3.0 and our CRISPRa-Suntag version rely on different recruitment mechanisms, which could account for some of the discrepancies observed. Interestingly, the Arabidopsis AD1, AD2 and AD9 could outperform 2xTAD and sometimes even VP64, displaying robust and stable activation in the endodermis. It is notable that the majority of ADs tested did not activate transcription. It is possible that the specific context of the dCas9-Suntag activation system and the use of tandem copies is incompatible with many ADs. The ADs DREB2, DOF1, and AvrXa10 were shown to outperform VP64 in the context of the Suntag activation system in *Arabidopsis*[33]. It would be interesting to compare these efficient ADs to address their robustness in different cell types or species.

### In planta guide RNA validation strategy using fluorescent promoter constructs

We have furthermore shown that current Cas9 systems cannot be expected to work reliably without previous testing of Cas9 expression and guide RNA functionality. A problem which is exacerbated the more components need to be expressed together. The need to assess Cas9 performance on a cell-type specific basis, required us to use novel benchmarking methods, using fluorescent promoter::reporter constructs as read-outs of activation potential. Despite being only a proxy for the activation of the endogenous gene, this way of testing is essentially without alternative for cell-type specific activation, since standard qPCR methods cannot provide this resolution, while other cell-type specific methods, such as in-situ hybridization or single cell sequencing, are far too onerous for testing purposes. Importantly, promoter::reporter fusions provide an information-rich read-out, allowing to judge timing of activation or consistency of expression ("patchiness" being indicative of potential silencing) and other unexpected spatial patterns of expression, all of which would be completely obfuscated in the averaging of tissue during RNA extraction for qPCR. The extensive use of fluorescent seed selection technology has also enabled us to pursue a novel way of testing guide RNA activity directly in the organism through stable transformation and analysis of T1 plants, again using fluorescent promoter::reporter constructs of genes-of-interest as read-outs. Expectedly, comparison with assays based on transient transformation of protoplasts demonstrates an increased predictability when guide RNAs are tested in stably transformed T1 lines. This is probably due to the high variability and very strong overexpression of both Cas9 and

guide RNAs in the protoplast system, as well as the particular cell state of protoplasts as compared to non-disturbed cells in their native developmental context. Our findings fit previous work reporting that rice protoplasts displayed a stronger activation potential than transgenic seedlings using identical gRNAs[27]. It is also important to point out that guide RNA performance should ideally be tested in the organ/cell-type of interest, using dCas9 under the control of cell-type specific promoter. In many cases, protoplast assays will require testing guide RNAs using constitutively overexpressed dCas9, since cell-type specific promoters might not be active in protoplasts. An important feature that determines the success of CRISPR activation is the establishment of a stable line expressing solely the activation system. Indeed, strong line-to-line variability in transcriptional activation can be observed when transforming *Arabidopsis* with vectors harboring both the activation system and the gRNA arrays, while very little variability was observed when supertransforming a stable CRISPRa line with a vector expressing just gRNAs.

### Stable flavonoid production in single root cell types

Using our rigorous testing strategy, we were able to go beyond current levels of gene activation and to induce expression of a functional suite of six endogenous genes in a specific cell-type, leading to accumulation of wild-type levels of flavonols in the endodermis. As mentioned before, the endodermis is a very good candidate for synthetic induction of flavonol biosynthesis by targeting its specific pathway genes, since it is a cell layer that has a highly active phenylpropanoid pathway for the production of lignin and suberin and should thus provide the necessary precursors for flavonol production. Yet, despite an efficient eventual flavonol accumulation, we were surprised that accumulation only occurred in fully differentiated endodermal cells, although Cas9 promoter activity starts already in early undifferentiated cells. Since an identical delay was observed when MYB12 was expressed with the same promoter, this observation is not due to the Cas9 activation system used. From what is known, an accumulation of flavonols co-incident with the formation of Casparian strips could have been expected. These lignified structures are the hallmarks of endodermal differentiation and were shown to be associated with endogenous lignin monomer production in the endodermis, which should provide necessary precursors for flavonols at this point[34]. The observed delay in flavonol production could indicate the presence of metabolic channeling, preventing access of PAL and C4H products for the flavonol biosynthetic enzymes. Indeed, knocking-out the substrate competitor for CHS in the lignin pathway was shown to cause strongly increased formation of flavonols[35]. Transferring our module of six "core" flavonol production genes into more cell layers by Cas9 activation is now very straightforward and could be used to test for sufficiency of this module in other cell layers, thus allowing a functional mapping of the metabolic environment of individual cell types.

Although we managed to specifically produce kaempferol and quercetin in the root endodermis of *myb12*, some flavonols could be detected in emerging lateral roots, suggesting active transport between cell layers in these tissues. Transport and modification of flavonols is of great relevance for understanding their function, but both aspects are still insufficiently understood[30]. In light of the importance of flavonoids in root rhizosphere interaction, nutrient uptake, plant defense, as well as human nutrition[36–38], it will be extremely interesting to produce flavonols by synthetic upregulation of their enzymatic pathway in a precise location, as this will allow to dissect their transport and modification routes with much greater precision. Finally, it will allow to conclude which role a metabolite plays in a given cell-type, tissue, or organ. In the case of lateral root formation, for example, accumulation of flavonols is observed in endodermis, pericycle, vasculature, as well as the developing primordia itself, making it difficult to understand in which cell layer flavonols are required and for what purpose.

### Use of Cas9 activation in metabolic re-programming

The differences between achieving metabolite production through expression of a transcription factor or a suite of metabolic enzymes is profound. From a fundamental perspective, achieving metabolite production exclusively by enzyme expression demonstrates that a given suite of enzymes is sufficient—within the context of a specific cell-type—to induce production of the metabolite. Expression of a transcription factor only demonstrates that the entire set of changes in the transcriptome of the target cell are sufficient to produce the metabolite, which can often represent hundreds of genes. Successful production of a much more targeted upregulation of a restricted enzyme set therefore, allows much more precise studies of the effects of a metabolite in vivo. Manipulation of metabolic pathways by transcription factors always comes with the caveat of potentially inducing much more complex metabolic changes. From a biotechnological perspective, metabolic engineering of flavonoids and the phenylpropanoid pathway in general is of great interest because of their widespread presence in many plant organs used for human consumption, as pigments in fruits, seeds, leaves and roots[39]. The ability to upregulate specific combinations of enzymes, representing pathways branches, or subsets, will be crucial for achieving controlled and effective production of defined sets of compounds. Cas9 activation is ideally suited for this purpose as it allows to rapidly test the consequences of various combinations of enzyme gene activations by simply transforming different arrays of guide RNAs. We predict that such rapid and precise manipulations of endogenous metabolic gene expression will also be extremely useful for rapidly improving the performance of pathways engineered through heterologous expression of transgenes. Impressive feats of metabolic engineering with transgenes have been achieved in recent years. One example is the formation of a caffeic acid cycle in stable tobacco plants by introduction of four fungal luciferase genes, generating autoluminescent plants. Interestingly, the efficiency of light production was clearly limited by the availability of endogenously produced caffeic acid[40]. Cas9 activation technology could be used to rapidly test for the best combination of endogenous enzyme activation that boosts the heterologous pathway while minimizing interference with other pathways. Another impressive recent example is the biosynthesis of a complex saponin, QS-21, by transfer of 20 genes into tobacco, albeit by transient expression[41]. Here, a limiting factor appeared to be the availability of isoleucine, enhanced production of which could again be conveniently attempted by upregulation of endogenous biosynthetic genes using Cas9 activation.

In summary, our work demonstrates the feasibility of metabolic pathway engineering in a developmental context, using Cas9 activation. Moreover, we provide a clear optimization protocol that should significantly increase the chances of successful, multiplexed gene activation in stable plant lines. While optimization of guide RNAs is currently a labor-intensive, empirical process, guide RNAs, once validated as effective, have a high probability of working in another context in the same species. Thus, once optimization has been done for a pathway of interest, the entry cost for other researchers will be significantly lower and could rapidly lead to the development of predictable, robust, and rapid manipulations for major metabolic pathways in plants using CRISPR activation technology.

## Methods

### Plant material and growth conditions

For all experiments, *Arabidopsis thaliana* (ecotype Columbia) was used. Seeds were surface-sterilized, sown on plates contained half-strength Murashige and Skoog (MS) + 0.8% Agar (Roth) medium, stratified at 4 °C and darkness for 2 days, and grown vertically in growth chambers at 21 °C at constant light (100 µE) for 5 days, unless stated otherwise. The mutant *myb12-1f* (Mehrtens et al., 2005), referred to as *myb12* was kindly provided by Prof. Ludwig-Müller. The following published constructs were introduced by floral dipping into

*myb12*: pPAL1::NLS-3xVenus (Andersen et al., 2021), pCH4::NLS-3xVenus (Andersen et al., 2021). The reporter lines pLOVE1::H2A-GFP, pGPAT2::NLS-GFP-GUS, and pGPAT3::NLS-GFP-GUS were gifted by Dr. Christiane Nawrath. The *tt7* mutant was provided by Prof. Yves Poirier.

## Cloning and plasmid construction

The Primestar polymerase (Takara), In-Fusion cloning kit (Takara), Gateway Cloning Technology (Invitrogen), and Golden Gate (Thermo Fisher) were used for generating constructs described thereafter, unless stated otherwise. All constructs were transformed by heat shock first into *Escherichia coli* Dh5α strain and then into *Agrobacterium tumefaciens* GV3101 strain, to be transformed into plants by floral dipping. All construct combinations and respective primers are listed in Supplementary Data 2.

## Generation of empty gateway entry vectors (dummy vectors)

The entry clone pAH128-pENTR-L4R1-EcorV-BglII-XbaI-BamHI containing a multicloning site in between L4 and R1 was generated by In-Fusion after PCR amplification of a L4R1 gateway vector backbone. The entry clone pAH166-pENTR-L1L2-KpnI-BglII-XhoI-EcoRI containing a multicloning site in between L1 and L2 was generated by In-Fusion after PCR amplification of a L1L2 gateway vector backbone. The entry clone pAH129-pENTR-R2L3-KpnI-BglII-XhoI-BamHI containing a multicloning site in between R2 and L3 was generated by In-Fusion after PCR amplification of a R2L3 gateway vector backbone.

## Generation of gRNA vectors

The gRNAs targeting the promoters used for transcriptional activation were cloned into intermediate vectors pRU41 (pAtU6-26), pRU42 (pAtU3), pRU43 (pAtU6-26), pRU44 (pAtU3), pRU45 (pAtU6-26) or pRU46 (pAtU3) (Ursache et al., 2021) and used for the dCas9-Suntag and dCas9-TV activations systems requiring standard gRNA scaffolds. For using the dCas9-Act2.0, the gRNA scaffold of pRU41, pRU42, and pRU43 was replaced by the MS2 stem loop containing gRNA scaffold of pYPQ131A2.0 (Addgene #99884) to generate pAH025-pRU41-2.0, pAH026-pRU42-2.0, and pAH027-pRU43-2.0, respectively. The AtU6-26 promoter and sgRNA backbone from the pEN-Chimera (Addgene #61432) were PCR amplified and cloned into pAH129 by In-Fusion to generate pAH208-pENTR-R2L3_1gRNAs. This vector was used for testing single gRNAs. The CcdB cassettes flanked by BsaI sites from the pSF463 (Addgene #167671), pSF278 (Addgene #167673), pSF464 (Addgene #167672), pSF279 (Addgene #167674) and pSF280 (Addgene #167675) were cloned into pAH129 to generate pAH209-pENTR-R2L3_2gRNAs, pAH210-pENTR-R2L3_3gRNAs, pAH211-pENTR-R2L3_4gRNAs, pAH212-pENTR-R2L3_5gRNAs and pAH213-pENTR-R2L3_6gRNAs, respectively and into an entry vector contain L4R1 sites to generate pAH203-pENTR-L4R1_2gRNAs, pAH204-pENTR-L4R1_3gRNAs, pAH205-pENTR-L4R1_4gRNAs, pAH206-pENTR-L4R1_5gRNAs and pAH207-pENTR-L4R1_6gRNAs. To stack gRNAs, the single gRNAs were multiplexed by Golden Gate, and the corresponding entry clones recombined by LR into expression vectors.

## Cloning of promoters

The LTPG20 promoter (1217 bp) was PCR amplified and cloned into pAH128 by In-Fusion using the BamHI restriction site to generate pAH008-pENTR-L4R1-pLTPG20.

## Generation of transcriptional reporters

pAH008-pENTR-L4R1-pLTPG20 and pENTR-L4R1-pPER03[42] were recombined with the pENTR-L1L2-NLS-3xmVenus into pFRm24GW to generate the transcriptional reporter for LTPG20 and PER03. The NLS-3xmVenus-tOCS cassette (2913 bp) was PCR amplified and cloned by In-fusion into an entry clone containing L1 and L2 gateway recombination sites to generate pAH343-pENTR-L1L2-NLS-3XVenus-tOCS-KpnI. The LOVE1 (1944bp), 4CL3 (1547 bp), CHI (2011bp), CHS

(1328 bp), F3H (1123 bp), FLS1 (1734 bp), TT7 (4517 bp), PAL1 (4075 bp) and C4H (4075 bp) promoters were PCR amplified and cloned by In-Fusion into the pAH343 to generate pAH345-pENTR-L1L2-pLOVE1-NLS-3XmVenus-tOCS, pAH346-pENTR-L1L2-p4CL3-NLS-3XmVenus-tOCS, pAH347-pENTR-L1L2-pCHI-NLS-3XmVenus-tOCS, pAH348-pENTR-L1L2-pCHS-NLS-3XmVenus-tOCS, pAH349-pENTR-L1L2-pF3H-NLS-3XmVenus-tOCS, pAH350-pENTR-L1L2-pTT7-NLS-3XmVenus-tOCS, pAH351-pENTR-L1L2-pFLS1-NLS-3XmVenus-tOCS, pAH617-pENTR-L1L2-pPAL1-NLS-3XmVenus-tOCS and pAH618-pENTR-L1L2-pC4H-NLS-3XmVenus-tOCS, respectively.

## The two module dCas9-Suntag CRISPR activation (CRISPRa) system

The constructs were adapted from the one reported in Papikian et al., 2019. The dCas9-Suntag-tOCS (6391 bp) was PCR amplified from the pEG302 22aa SunTag VP64 nog (Addgene #120249) and cloned by In-Fusion into the pAH128 to generate the pAH131-pENTR-L4R1-XbaI-dCas9-Suntag-tOCS. The LTPG20 promoter (1217 bp) was PCR amplified and cloned by In-Fusion into the pAH131 to generate the pAH132-pENTR-L4R1-pLTPG20-dCas9-Suntag-tOCS. The PER03 promoter (2026 bp) was PCR amplified and cloned by In-Fusion into the pAH131 to generate the pAH133-pENTR-L4R1-pPER03-dCas9-Suntag-tOCS. The ScFv-GCN4 (899 bp) and VP64-tNOS (784 bp) were PCR amplified from the pEG302 22aa SunTag VP64 nog (Addgene #120249) to remove the superfolder GFP and cloned by In-Fusion into pAH166 using the EcoRI site to generate the pAH134-pENTR-L1L2-KpnI-BglII-XhoI-EcoRI-scFv-GCN4-VP64-tNOS. The LTPG20 promoter (1217 bp) was amplified by PCR and cloned by In-Fusion into the pAH134 to generate the pAH135-pENTR-L1L2-pLTPG20-scFv-GCN4-VP64-tNOS. The PER03 promoter (2026 bp) was PCR amplified and cloned by In-Fusion into the pAH134 to generate the pAH136-pENTR-L1L2-pPER03-scFv-GCN4-VP64-tNOS.

## The two-in-one dCas9-Suntag CRISPRa system

The constructs were adapted from the one reported in Papikian et al., 2019. The ScFv-GCN4-VP64 cassette was codon optimized for *Arabidopsis* and synthetized by Genscript. VP64 was flanked by XhoI restriction sites to be able to swap the activation domain. The synthetized fragment was cloned into a gateway vector containing L4R1 recombination sites to generate the pAH291-pENTR-L4R1-ScFv-VP64-KpnI-XhoI-XbaI. The dCas9-Suntag-tOCS (6391 bp) was PCR amplified from the pEG302 22aa SunTag VP64 nog (Addgene #120249) and cloned by In-Fusion using the XbaI restriction site into the pAH291 to generate pAH292-pENTR-L4R1-dCas9-Suntag-VP64-XbaI-XhoI-KpnI. The VP64 activation domain in pAH292 was replaced by 2xTAD that was PCR amplified from pYPQ-dpcoCas9-Act3.0 (Addgene #158408) and cloned by In-fusion to generate pAH293-pENTR-L4R1-dCas9-Suntag-2xTAD-XbaI-XhoI-KpnI. The PER03 (2026 bp), UBQ10 (1986bp), ZmUbi (1998bp), GPAT3 (2223 bp) and PEP (2235 bp) promoters were PCR amplified and cloned by In-Fusion into the pAH292 to generate pAH294-pENTR-L4R1-pPER03-dCas9-Suntag-tOCS-pPER03-ScFv-VP64, pAH334-pENTR-L4R1-pAtUBQ10L-dCas9-Suntag-tOCS-pZmUbi-ScFv-VP64, pAH337-pENTR-L4R1-pGPAT3-dCas9-Suntag-tOCS-pGPAT3-ScFv-VP64 and pAH653-pENTR-L4R1-pPEP-dCas9-Suntag-tOCS-pPEP-ScFv-VP64, respectively. The PER03 (2026 bp) promoter was cloned by In-Fusion into the pAH293 to generate the pAH315-pENTR-L4R1-pPER03-dCas9-Suntag-tOCS-pPER03-ScFv-2xTAD. To generate the Suntag activation system variants using alternative activation domains, pAH293-pENTR-L4R1-dCas9-Suntag-2xTAD-XbaI-XhoI-KpnI was digested with XbaI, and 2xTAD was replaced with a tandem copy of AD1 to AD15 (Supplementary Data 1).

## Vectors for testing individual gRNAs efficiency

For testing gRNAs efficiency in stable independent T1 seedlings, pAH315-pENTR-L4R1-pPER03-dCas9-Suntag-tOCS-pPER03-ScFv-

2xTAD was recombined with the transcriptional reporters of 4CL3(pAH346), CHI(pAH347), CHS (pAH348), F3H (pAH349), TT7 (pAH350), FLS1 (pAH351), PAL1 (pAH617) or C4H (pAH618) and the respective entry clone harboring the individual gRNA in pAH208-pENTR-R2L3_1gRNAs into the triple gateway destination vector pFRm34GW containing a Fast-red seed selection marker. For testing gRNAs efficiency in protoplasts, pAH335-pENTR-L4R1-pAtUBQ10L-dCas9-Suntag-tOCS-pZmUbi-ScFv-2xTAD was recombined with the transcriptional reporters of 4CL3(pAH346), CHI(pAH347), CHS (pAH348), F3H (pAH349), TT7 (pAH350) or FLS1 (pAH351) and the respective entry clone harboring the individual gRNA in pAH208-pENTR-R2L3_1gRNAs into the triple gateway destination vector pB7m34GW0-pUBQ10L-NLS-tdTomato-tUBQ10 containing a ubiquitously expressed NLS-tdTomato marker on the vector backbone.

### The dCas9-Act2.0 CRISPRa system
The constructs were adapted from the one reported in Lowder et al., 2017. The dCas9-Act2.0 cassette was PCR amplified and cloned into a pDONR221 to generate pAH023-pENTR-L1L2-dCas9-VP64-T2A-MS2-VP64.

### The dCas9-TV CRISPRa system
The constructs were adapted from the ones reported in Li et al., 2019. The dCas9-TV cassette was PCR amplified from the HBT-dCas9-TV kindly provided by Prof. Jian-Feng Li and cloned into a pDONR221 to generate pAH024-pENTR-L1L2-dCAS9-TV.

### RNA extraction and qRT-PCR
For RNA extraction, seedlings were grown for 5 days on half strength MS on mesh to facilitate root cutting. Approximately 100 mg roots (at least 50 seedlings) were cut and frozen in liquid nitrogen and stored at −80 °C. Total RNA was extracted using the ReliaPrep RNA Tissue Miniprep Kit (Promega). Reverse transcription was carried out with PrimeScript RT Master Mix (Takara). All steps were done as indicated in the manufacturer's protocols. The qPCR was performed on an Applied Biosystems QuantStudio3 thermocycler using a MESA BLUE SYBR Green kit (Eurogentech). Transcripts are normalized to Clathrin adaptor complexes medium subunit family protein ADAPTOR PROTEIN-4 MU-ADAPTIN, AP4M (AT4G24550) expression. Experiments were carried out in triplicate. All primer sets are indicated in Supplementary Data 2.

### Protoplast isolation and transformation
Protoplasts were isolated from *ler-0* cell cultures. Cell cultures were maintained in liquid medium containing MS (Duchefa M0222) 4.43 g/l, 3% sucrose, 500 µg/l NAA, 50 µg/l kinetin at pH 5.6. For protoplast isolation, 25 ml of a 2-3 days old cell culture were spinned down at 200 g for 5 min. The supernatant was removed, and cells were resuspended in 50 ml GM buffer containing cell wall digestion enzymes (MS (Duchefa M0222) 4.43 g/l, 0.17 M Glucose, 0.17 M Mannitol, pH5.5, 1% Cellulase, 0.2% Macerozyme, filter sterilized (0.22 µm). Cells were transferred to a large petri dish and incubated for 4–6 h in the dark at room temperature with gentle shaking (40–50 rpm). Protoplasts were filtered through a cell strainer (100 µm) spinned down for 5 min at 100 g, supernatant was removed, and cells were washed with 25 ml GM buffer. Cells were spinned down and resuspended in 12 ml S buffer (MS (Duchefa M0222) 4.43 g/l, 0.28 M Sucrose, pH 5.5, filter sterilized (0.22 µm) in a 15 ml falcon. The protoplasts were recovered in the upper phase of the sucrose gradient after centrifugation at $100 \times g$ for 10 min. Protoplasts were transformed by PEG mediated transformation (25% PEG 6000, 0.45 M Mannitol, 0.1 M Ca(NO₃)₂, pH9, filter sterilized 0.45 µm). Briefly, $5 \times 10^5$ protoplasts in 50 µl were transformed with 12 µg vector DNA in 15 µl distilled water. 150 µl PEG solution was added immediately and mixed by gentle flicking of the Eppendorf tube. Cells were incubated for 45 min in the dark. PEG was

washed by adding 1 ml of 0.275 M Ca(NO₃)₂ in two steps of 0.5 ml. Cells were spinned down at 100 g for 5 min, resuspended in 0.5 ml GM buffer, and incubated overnight in the dark. Protoplasts were imaged 24 h after transformation.

### Fluorescence Microscopy
Confocal laser-scanning microscopy images were obtained using a Zeiss LSM 880 (with Zen 2.1 SP3 Black edition). Seedlings were fixed using PFA and cleared in Clearsee as described earlier (Ursache et al., 2018). The following excitation and detection windows were used: Calcofluor White 405 nm, 430–485 nm, mVenus/GFP 488 nm, 500–530 nm, tdTomato, 561 nm, 590–650 nm. For quantifications of gRNA efficiency in protoplasts, the ratio between the YFP signal over RFP signal was calculated. This ratiometric quantification allowed to reduce variability inherent to transient protoplast transformation. The flavonols kaempferol and quercetin were visualized in vivo using Diphenylboric acid 2-aminoethyl ester (DPBA, 2-Aminoethyl diphenylborinate by Sigma Aldrich; Catalog number D9754). Seedlings were stained in darkness in 0.25% w/v DPBA with 0.06% Triton-X (v/v) in water for 20 min, with a 20 min wash in water. Seedlings were mounted on microscope slides in water. Roots were imaged using the 458 laser at 30% laser power with an emission spectrum of 472–500 for the kaempferol channel and of 585–619 for the quercetin channel. All microscope settings were kept identical between each sample. Overview images of fixed whole roots of 5-days-old seedlings were acquired on a Leica DM6B THUNDER (LasX 3.7.4.23463). The following excitation and detection windows were used: Calcofluor White 375-435, 450–490 nm, mVenus/GFP 450–490 nm, 500–550 nm. For screening lines based on their transcriptional activation potential, 5-days-old seedlings were analyzed in vivo on a Zeiss Axiozoom.V16 stereomicroscope. Fluorescent seeds carrying FastGreen or FastRed cassettes were selected using a Leica MZ16FA Fluorescence Stereomicroscope. The filters used for different colours are as follows: DSR (LEICA 10447227) for FastRed; GFP3 (LEICA 10447217) for FastGreen.

### Image analysis
All images were processed identically using FIJI to allow comparison between samples. Some fixed whole roots have been straightened using the "straighten" tool.

### Statistics and reproducibility
All experiments were carried out in replicates with similar outcomes. No statistical methods were used to predetermine sample size. No data was excluded from the analyses. The experiments were not randomized, and investigators were not blinded to allocation during experiments and outcome assessment.

### Reporting summary
Further information on research design is available in the Nature Portfolio Reporting Summary linked to this article.

## Data availability
The source data for all main and Supplementary Figs. are provided as a Source Data file. All other data that support the findings of this study are available from the corresponding authors upon reasonable request. Source data are provided with this paper.

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

## Acknowledgements

This work was supported by an EMBO Postdoctoral Fellowship (grant number ALTF 1153-2019) and a Marie Curie Standard European Fellowships (EF-ST, grant number 892018) to AH and an ERC Advanced Grant (ROOBABA, 101020794) and a Swiss National Science Foundation (SNSF) grant (10002702) to NG.

## Author contributions

A.H. and N.G. designed the experiments and wrote the manuscript. A.H. carried out all the experiments with contribution of V.D.T. The protoplasting experiments were performed by A.H. and L.H. The list of activation domains to be validated in Arabidopsis was provided and curated by N.M. and L.C.S.

## Competing interests

The authors declare no competing interests.
