## [Transparent Peer Review file · Nature Communications]

Efficient, cell-type-specific production of flavonols by multiplexed CRISPR activation of a suite of metabolic enzymes

Corresponding Author: Professor Niko Geldner

Version 0:

Reviewer comments:

Reviewer #1

(Remarks to the Author)

The authors largely addressed all my questions with reasonable arguments. In the revision, they further made a comparison between the protoplast assay and the stable line assay, which showed the reliability of their preferred assay in stable lines. Given the level of innovation presented in this study and the interesting discoveries made (anticipated or surprising), I feel this work in principle can be published in Nature Communications. Here I have a few minor points for the authors to consider to enhance the manuscript further.

1. The use of CRISPRa for metabolic engineering is a highlight of this paper. The authors successfully demonstrated that simultaneous activation of the metabolic pathway enzyme genes in a myb12 mutant led to the accumulation of flavonols at the WT level. However, a major goal of metabolic engineering is to obtain a higher amount of the given metabolic in the control (WT) conditions, whether using a transient expression system (as done in Plant Biotechnol J. 2022 Aug; 20(8): 1578–1590) or a stable expression system (as done in this paper). I asked the authors whether they tried their CRISPRa system in the WT background because I was curious to know whether that can lead to higher flavonol accumulation than WT. It may not work given the levels of CRISPRa and the complexity of the metabolic regulation or silencing as the authors discovered. However, this is an obvious question that many readers may ask when reading this paper. So, I'd prefer a direct answer to my question (yes or no). If the authors tried it in the WT background but didn't observe a higher accumulation of target metabolite, they should report it even if the data is negative (but informative).
2. Line 77-80: Some revision is needed for more clarity. Based on the reference cited and description, it was meant to talk about CRISPR-Combo (not CRISPR-Act3.0). In that case, I suggest mentioning CRISPR-Combo directly and citing the original paper: PMID: 35596077.
3. Supplementary Fig. 7. The panel "C" in the figure needs to be spelled out in the legend, which is on protoplast assay.
4. Lines 308-310: I liked that the authors discussed the CRISPRa results in a more cautious way, bringing up the specific context. In fact, in the CRISPR-Ac3.0 paper, it was shown 2xTAD outperformed VP64 when CRISPRa was done via MS2-MCP based recruitment of Suntag (Fig. 2c). So, context definitely matters, and this result may be worth mentioning to further support it. Also, different Cas9 versions (different codon optimization and with or without introns) may account for the difference in CRISPRa as well. I think a pco-dCas9 was used in both dCas9-TV and CRISPR-Act2.0 systems and this Cas9 has an intron. In this manuscript, the authors used the Cas9 from the Steve Jacobson's lab, which is a different Cas9 version. So, I suspect this difference (of using different Cas9 clones) may also have an effect.
5. To be consistent, I suggest replacing "Cas9 activation" with "CRISPR activation" or "CRISPRa". It is clearer this way.

Reviewer #2

(Remarks to the Author)

In this manuscript, the authors demonstrated the utilization of multiplexed CRISPR activation to co-activate up to six genes

in the endodermis or epidermis of Arabidopsis roots, enabling cell-type-specific production of flavonols. While the study presents some interesting findings, in my view, its technical advancement, practical significance, and general appeal are limited. Consequently, this work seems more appropriate for a plant-focused journal like Plant Biotechnology Journal rather than Nature Communications.

Major comments:

1. Limited technical novelty: The gene activation system employed in this research is a modified dCas9-Suntag system regulated by a cell-type-specific promoter. As such, I perceive the technical novelty of this work to be rather low.

2. Doubts about the practical relevance: I remain unconvinced by the efficiency and practical significance of the cell-type-specific multiplexed CRISPR system described in this study. Firstly, the process to achieve cell-type-specific multigene co-activation is extremely labor-intensive. The authors had to screen gRNAs targeting individual genes in transgenic plants, utilized fluorescent promoter constructs to measure cell-type-specific gene activation outcomes, and relied on the myb12 mutant to verify their concept. In contrast, directly using a cell-type-specific promoter to drive the expression of multiple biosynthetic genes appears to be a more straightforward and less labor-intensive approach. After all, constructing and co-expressing up to ten transgene cassettes is not overly difficult nowadays, see PMID: 29901840. Moreover, the production of flavonoids in single root cell types was affected by active metabolic transport to adjacent regions. I would expect to see evidence that this cell-type-specific multiplexed CRISPR activation can truly address a fundamental question in plant biology or re-engineer a metabolic pathway with practical importance and impact.

Reviewer #3

(Remarks to the Author)

In the revised manuscript, the authors have suitably addressed my comments on the previous submission of their study. Their additional demonstration of Cas9 activation of flavanol production in root cap/epidermal cells further demonstrates the applicability of their system. Furthermore, the new comparison of stable transgenic vs protoplast assays is valuable as it emphasises the strengths and weaknesses of both approaches (speed/throughput vs accuracy) that need to be considered.

I would reiterate my prior comments that, overall, this is a highly valuable study that significantly advances the ability to more precisely manipulate the activity of multiple genes in a cell type specific manner in plants. The work provides several novel contributions that extend beyond prior publications. The authors' use of visual reporters provide valuable insights into the CRISPRa activity. Their component testing provides significant and unanticipated new insights into those that are most effective for in vivo CRISPRa, and they demonstrate the capabilities of this approach through sophisticated multiplexed pathway manipulation. While at the outset one might assume that cell type specific expression of CRISPRa components should of course achieve cell specific gene activation, there are numerous potential points of failure. The authors' systematic work to overcome a variety of obstacles, and description of cloning and screening approaches for effective optimisation, provide a valuable framework that I'm sure will be broadly adopted to enable far more sophisticated dissection of cell type specific properties and pathways.

REVIEWERS' COMMENTS

Reviewer #1 (Remarks to the Author):

The authors largely addressed all my questions with reasonable arguments. In the revision, they further made a comparison between the protoplast assay and the stable line assay, which showed the reliability of their preferred assay in stable lines. Given the level of innovation presented in this study and the interesting discoveries made (anticipated or surprising), I feel this work in principle can be published in Nature Communications. Here I have a few minor points for the authors to consider to enhance the manuscript further.

1. The use of CRISPRa for metabolic engineering is a highlight of this paper. The authors successfully demonstrated that simultaneous activation of the metabolic pathway enzyme genes in a myb12 mutant led to the accumulation of flavonols at the WT level. However, a major goal of metabolic engineering is to obtain a higher amount of the given metabolic in the control (WT) conditions, whether using a transient expression system (as done in Plant Biotechnol J. 2022 Aug; 20(8): 1578–1590) or a stable expression system (as done in this paper). I asked the authors whether they tried their CRISPRa system in the WT background because I was curious to know whether that can lead to higher flavonol accumulation than WT. It may not work given the levels of CRISPRa and the complexity of the metabolic regulation or silencing as the authors discovered. However, this is an obvious question that many readers may ask when reading this paper. So, I'd prefer a direct answer to my question (yes or no). If the authors tried it in the WT background but didn't observe a higher accumulation of target metabolite, they should report it even if the data is negative (but informative).

REPLY: We thank the reviewer for raising this concern. We did not try to increase flavonol accumulation in a WT background.

2. Line 77-80: Some revision is needed for more clarity. Based on the reference cited and description, it was meant to talk about CRISPR-Combo (not CRISPR-Act3.0). In that case, I suggest mentioning CRISPR-Combo directly and citing the original paper: PMID: 35596077.

REPLY: The reviewer raised a valid point. CRISPR-Act3.0 was modified to CRISPR-Combo in Line 74-75

3. Supplementary Fig. 7. The panel “C” in the figure needs to be spelled out in the legend, which is on protoplast assay.

REPLY: The panel (c) has now been spelled out in the figure legend

4. Lines 308-310: I liked that the authors discussed the CRISPRa results in a more cautious way, bringing up the specific context. In fact, in the CRISPR-Ac3.0 paper, it was shown 2xTAD outperformed VP64 when CRISPRa was done via MS2-MCP based recruitment of Suntag (Fig. 2c). So, context definitely matters, and this result may be worth mentioning to further support it.

REPLY: A statement has been added to the discussion. Line 302-304

Also, different Cas9 versions (different codon optimization and with or without introns) may account for the difference in CRISPRa as well. I think a pco-dCas9 was used in both dCas9-TV and CRISPR-Act2.0 systems and this Cas9 has an intron. In this manuscript, the authors used the Cas9 from the Steve Jacobson's lab, which is a different Cas9 version. So, I suspect this difference (of using different Cas9 clones) may also have an effect.

REPLY: We fully agree with the reviewer's comment

5. To be consistent, I suggest replacing "Cas9 activation" with "CRISPR activation" or "CRISPRa". It is clearer this way.

REPLY: This has been adapted in the manuscript

Reviewer #2 (Remarks to the Author):

In this manuscript, the authors demonstrated the utilization of multiplexed CRISPR activation to co-activate up to six genes in the endodermis or epidermis of Arabidopsis roots, enabling cell-type-specific production of flavonols. While the study presents some interesting findings, in my view, its technical advancement, practical significance, and general appeal are limited. Consequently, this work seems more appropriate for a plant-focused journal like Plant Biotechnology Journal rather than Nature Communications.

Major comments:

1. Limited technical novelty: The gene activation system employed in this research is a modified dCas9-Suntag system regulated by a cell-type-specific promoter. As such, I perceive the technical novelty of this work to be rather low.

REPLY: Our use of transcriptional reporters has enabled us to demonstrate successful, cell-type specific re-wiring of expression of genes expressed in different cell-type, something which is essential for synthetic biology purposes using CRISPR activation. We maintain that this is a clear novelty in the field.

2. Doubts about the practical relevance: I remain unconvinced by the efficiency and practical significance of the cell-type-specific multiplexed CRISPR system described in this study. Firstly, the process to achieve cell-type-specific multigene co-activation is extremely labor-intensive. The authors had to screen gRNAs targeting individual genes in transgenic plants, utilized fluorescent promoter constructs to measure cell-type-specific gene activation outcomes, and relied on the myb12 mutant to verify their concept. In contrast, directly using a cell-type-specific promoter to drive the expression of multiple biosynthetic genes appears to be a more straightforward and less labor-intensive approach. After all, constructing and co-expressing up to ten transgene cassettes is not overly difficult nowadays, see PMID: 29901840.

REPLY: Whilst co-expression of multiple transgenes is becoming quite feasible these days, reports of achieving functional, stable and cell-type expression of several transgenes are still extremely rare. The cited paper certainly did not demonstrate this. The aim of our study was to test the potential and limits of CRISPR activation systems in this context. We maintain that CRISPR activation has significant advantages over the generation of multiple transgenes.

Moreover, the production of flavonoids in single root cell types was affected by active metabolic transport to adjacent regions. I would expect to see evidence that this cell-type-specific multiplexed CRISPR activation can truly address a fundamental question in plant biology or re-engineer a metabolic pathway with practical importance and impact.

REPLY: Our paper represents a proof of concept of the possible uses of CRISPR activation technology for answering fundamental questions.

Reviewer #3 (Remarks to the Author):

In the revised manuscript, the authors have suitably addressed my comments on the previous submission of their study. Their additional demonstration of Cas9 activation of flavanol production in root cap/epidermal cells further demonstrates the applicability of their system. Furthermore, the new comparison of stable transgenic vs protoplast assays is valuable as it emphasises the strengths and weaknesses of both approaches (speed/throughput vs accuracy) that need to be considered.

I would reiterate my prior comments that, overall, this is a highly valuable study that

significant advances the ability to more precisely manipulate the activity of multiple genes in a cell type specific manner in plants. The work provides several novel contributions that extend beyond prior publications. The authors' use of visual reporters provide valuable insights into the CRISPRa activity. Their component testing provides significant and unanticipated new insights into those that are most effective for in vivo CRISPRa, and they demonstrate the capabilities of this approach through sophisticated multiplexed pathway manipulation. While at the outset one might assume that cell type specific expression of CRISPRa components should of course achieve cell specific gene activation, there are numerous potential points of failure. The authors' systematic work to overcome a variety of obstacles, and description of cloning and screening approaches for effective optimisation, provide a valuable framework that I'm sure will be broadly adopted to enable far more sophisticated dissection of cell type specific properties and pathways.

REPLY: We thank the reviewer for their positive feedback on the revised manuscript.